# Decoupling structural molecular dynamics from excited state lifetimes using few-femtosecond ultraviolet resonant dispersive waves

Sebastian L. Jackson [1,4], Andrew W. Prentice [2], Lauren Bertram [3], Lewis Hutton [3], Nikoleta Kotsina [1], Christian Brahms [1], Chris Sparling [1], John C. Travers [1], Adam Kirrander [3], Martin J. Paterson [2] & Dave Townsend [1,2] ✉

Optical sources exploiting resonant dispersive wave (RDW) emission are set to revolutionize ultrafast science. We demonstrate this approach by investigating excited state dynamics in morpholine using time-resolved photoelectron imaging. Excitation at 250 nm was achieved via RDW emission inside a helium-filled capillary fibre which, when combined with a short 800 nm probe, realized an instrument response of just $11 \pm 2$ fs. Two pathways initiate N−H bond fission: an extremely fast (<10 fs) process and a frustrated mechanism (380 fs) with hindered electronic ground state access. Photoelectron angular distributions also indicate average molecular geometry evolving on an intermediate (~100 fs) timescale. This clean distinction between population lifetimes and structural dynamics is enabled by the excellent temporal resolution inherent in RDW-based sources. Electronic structure and nonadiabatic surface hopping calculations support our data interpretation, and the synergy between experiment and theory is vital for developing a complete mechanistic picture.

Soliton dynamics in gas-filled hollow capillary fibers (HCFs) offer a robust, low-cost, and compact source of ultrashort (<5 fs) and continuously tunable deep-ultraviolet (DUV) and vacuum-ultraviolet (VUV) pulses of light[1]. Upon coupling an intense infrared (IR) laser input into a gas-filled HCF, the interplay between optical nonlinearity and group-velocity dispersion can lead to spectral broadening and pulse self-compression to sub-cycle duration. The self-compressed pulse may then undergo soliton fission, leading to a phase-matched, resonant transfer of energy to the DUV or VUV spectral regions in a process known as resonant dispersive wave (RDW) emission. The central wavelength of the RDW can be tuned simply by varying the pressure of the fill gas inside the HCF[2,3]. Because it can efficiently create tuneable VUV/DUV pulses with a few-femtosecond duration, the RDW phenomenon is rapidly emerging as a novel technique for ultrafast spectroscopy, which will play a leading role in next-generation ultrafast science. Such time-resolved investigations are relevant to a broad range of sub-disciplines within chemistry, physics, biology, and materials science, and the impact of RDW-based light sources will therefore be far-reaching. Studies with RDW-based ultraviolet light sources have, however, so far mostly reported on pulse characterization[4–9] or proof-of-principle experiments[6,10,11]. With the exception of one optical pump–probe experiment[12], ultraviolet

[1]Institute of Photonics & Quantum Sciences, Heriot-Watt University, Edinburgh, UK. [2]Institute of Chemical Sciences, Heriot-Watt University, Edinburgh, UK. [3]Physical and Theoretical Chemistry Laboratory, Department of Chemistry, University of Oxford, Oxford, UK. [4]Present address: Elettra - Sincrotrone Trieste S.C.p.A., Basovizza, Trieste, Italy. ✉e-mail: d.townsend@hw.ac.uk

**Fig. 1 | Summary of spectral parameters (see "Methods" section for more details). a** Intensity-normalized traces of the broadband IR input spectrum coupled into a (0.75 m) He-filled HCF used for RDW generation, and the corresponding DUV output spectrum centered at 250 nm. **b** Energy integrated photoelectron data obtained from non-resonant DUV-IR photoionization of 1,3-butadiene using the DUV pump centered at 250 nm, plus a separate portion of the IR pulse as a probe. A Gaussian fit to these data (solid red line) is also overlaid. Uncertainties in the cross-correlation FWHM and error bars are 1σ values.

dispersive waves generated in HCFs have yet to be applied to an open scientific question.

Here, we interrogate excess energy redistribution dynamics in the photoexcited electronic states of morpholine molecules using the time-resolved photoelectron imaging (TRPEI) technique. The experiment exploits few-femtosecond DUV pump pulses generated by RDW emission in conjunction with a short (~10 fs) multiphoton IR probe. The excellent time resolution afforded by this approach allows us to identify two distinct pathways governing molecular photofragmentation. As expanded upon later, these have been inferred previously from frequency domain measurements interrogating the vibrationally resolved distribution of organic radical products, but never observed directly. Moreover, the structural rearrangement of the molecule and the population evolution after photoexcitation are decoupled, with the structural dynamics occurring on an intermediate timescale between the fast and slow components of the population transfer. We support our interpretation with high-level electronic structure and trajectory surface-hopping simulations. Our work provides an important demonstration of the capabilities afforded by RDW emission as a light source for ultrafast spectroscopy and, furthermore, showcases the current state-of-the-art synergy between advanced quantum chemistry calculations and experimental measurements.

We have previously demonstrated the direct integration of an HCF output into the vacuum assembly of a TRPEI spectrometer, generating extremely short (approx. 5 fs) pump pulses in the deep-ultraviolet spectral region via RDW emission[6]. By combining these pulses with a temporally short 800 nm multiphoton probe, we were able to realize a Gaussian pump-probe instrument response function (or cross-correlation) on the order of just 10 fs full-width at half-maximum (FWHM). A full technical overview of the experimental infrastructure and optical beam paths may be found elsewhere[6,13], with a shorter summary (including all relevant optical pulse parameters) also provided in the "Methods" section. By implementing several upgrades to our earlier prototype setup, we have now significantly extended the usable lifetime of a single section of HCF, improved the shot-to-shot stability, and increased the brightness of the output pulses (now up to 10% IR-to-DUV conversion efficiency). This was achieved by adding an additional stage of differential pumping to permit the use of a high-pressure helium fill gas inside the HCF (up to 12 bar) in place of low-pressure argon (300–600 mbar); better optimizing the launch conditions of the IR driving pulse into the HCF; and refining strategies to ensure the HCF is held as straight as possible to avoid bend losses[14].

Figure 1 provides a summary of the RDW spectral output now generated in the DUV at a central wavelength of 250 nm, along with that of the short (10 fs) IR input driving pulse. Also shown is the cross-correlation obtained when using these pulses in combination to induce non-resonant ionization of the 1,3-butadiene molecule via absorption of one DUV photon and three IR photons (a process denoted as $1+3'$). A Gaussian fit to these data yields a FWHM of just $11\pm2$ fs. This is reproducible day-to-day and is consistent under small experimental shifts in the temporal sampling with respect to the true zero pump–probe delay position.

The significant upgrades outlined above have now, for the first time, enabled extended RDW-TRPEI measurements investigating the energy redistribution mechanisms operating in the photoexcited electronic states of molecules. The TRPEI approach is a powerful pump–probe technique which offers highly differential time-, energy- and angle-resolved information in a single set of experimental measurements. This permits detailed tracking of initially excited electronic states evolving toward various photoproducts in real time, yielding deep insight into the underlying photophysics[15–17]. The use of TRPEI coupled with an HCF-based RDW output represents an important advance in the field of excited-state molecular dynamics, as our experiment significantly improves the observation of the earliest time events after absorption of a DUV photon. This exploits the extremely short RDW pulse duration and opens interesting avenues for novel spectroscopic investigations in the ultrafast domain.

Morpholine (schematic structure in Fig. 2) is a saturated cyclic secondary amine that provides an excellent starting model for investigating the fundamental photophysics of the N−H chemical bond, which is ubiquitous throughout nature. The molecule photodissociates after absorbing DUV radiation below approx. 255 nm following excitation to its lowest-lying singlet electronic state ($S_1$), which is predominantly of 3s Rydberg character in the vertical Franck−Condon region[18,19]. Note here that we refer to adiabatic states throughout. Oliver et al. have previously studied the photodissociation of morpholine using H (Rydberg) atom photofragment translational spectroscopy over a range of UV absorption wavelengths (193–250 nm)[19]. At excitation wavelengths longer than 220 nm, two distinct H atom elimination pathways were observed, with both leading to the formation of concomitant ground ($\tilde{X}$) state morpholinyl radicals: (i) a high kinetic energy release channel exhibiting significant recoil anisotropy; and (ii) a low kinetic energy release channel that suggests a slower, more "frustrated" dissociation mechanism. The

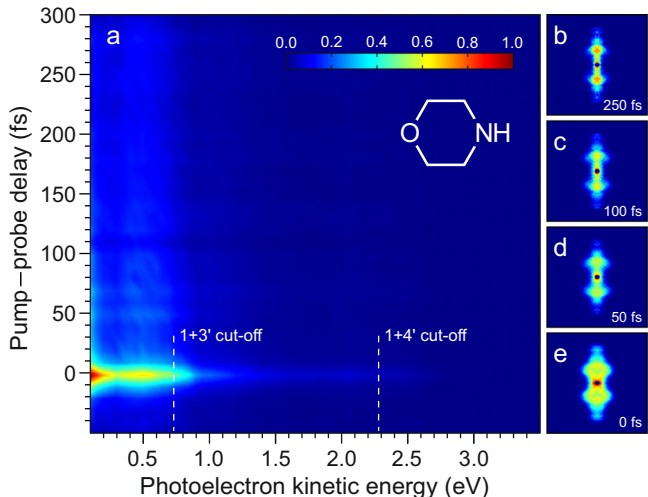

**Fig. 2 | Photoelectron imaging data. a** Time-resolved photoelectron spectrum of morpholine (schematic structure inset) following 250/800 nm pump/probe ionization. **b–e** photoelectron images at selected timesteps (pump and probe laser polarizations are vertical with respect to the figure). These have been 4-fold symmetrized (i.e., the raw data in four individual quadrants defined by horizontal and vertical axes passing though the image center are averaged together – which is valid given the symmetry of the photoelectron angular distribution described by Eq. 2). A small mask of radius 8 pixels radius has also been applied to images centers and the intensity then rescaled to highlight how the photoelectron angular distribution varies over time – which is otherwise obscured by the changes in total signal intensity.

former was attributed to direct passage through a conical intersection (CI) connecting the first excited singlet state of morpholine and the $S_0$ ground state, while the latter was assigned to a process where the system takes considerably longer to achieve the prerequisite geometry for efficient internal conversion. In such a frequency-resolved measurement, however, only the asymptotic photoproduct limit is probed, and these pathways cannot be observed directly. As highlighted elsewhere[20], a combination of time- and frequency-resolved measurements is often necessary to fully interpret the rich and complex dynamics operating in the excited states of polyatomic molecules. Cleavage of the morpholine N–H bond during the photodissociation is expected to proceed extremely rapidly in some cases (-10 fs) and via a large change in the electronic character. The combination of exquisite time resolution afforded by RDW emission and the highly differential information contained in the photoelectron angular distribution (PAD) obtained from the TRPEI measurement makes our experiment extremely well-suited to resolving the two distinct pathways in real time.

## Results

The popular velocity-map imaging (VMI) technique[21] was used to acquire raw TRPEI data. This comprised a set of 2D projection images of the full 3D-PAD at each pump–probe delay, recorded using a microchannel plate and phosphor screen assembly in conjunction with a CCD camera (see "Methods" section for more details). Polar basis-set expansion (pBASEX)[22] was then used to process this data, permitting quantitative analysis of the excited-state dynamics operating in morpholine following DUV excitation at around 250 nm using our RDW source. At this pump wavelength, excitation of the 3s ← $n$ Rydberg transition induces hybridization at the N atom center from $sp^3$ to $sp^2$, bringing about an associated change in the localized equilibrium geometry from pyramidal to planar. The initially excited state also exhibits increased σ* character as the N–H bond stretches – a common feature of many hydrocarbon species containing N, O, or S heteroatoms[23–26]. We therefore label the initially excited state $S_1$(3s/nσ*) to reflect its mixed Rydberg/valence composition over the full

range of nuclear geometries that may be sampled following photoexcitation. Figure 2 presents a time-dependent photoelectron spectrum obtained using our DUV-IR pump–probe ionization scheme. Raw image data at selected pump–probe delay times Δ*t* are also included, revealing a marked variation in PAD structure over time – as will be a focus of detailed discussion later. The main contribution to the photoelectron spectrum is a broad band at kinetic energies below a cut-off close to 0.7 eV. Based on the known (vertical) ionization potential of morpholine (8.88–8.91 eV)[27,28], this may be attributed to 1+3′ ionization of the $S_1$(3s/nσ*) state populated in the initial optical excitation. A weaker tail extending to higher photoelectron kinetic energies is assigned to a minor contribution from 1+4′ ionization. This distinct multiphoton behavior (and the lack of any signal extending beyond this second limit) indicates that strong-field ionization is not a significant factor in the experiment. Upon varying the 800 nm probe intensity, no appreciable change was observed in the relative strength of the ionization signals obtained close to Δ*t* = 0 and at longer pump–probe delay times. This indicates that two-color non-resonant ionization is not a significant factor in our measurements.

A global fitting routine was used to extract quantitative information from the data presented in Fig. 2 and investigate the time-dependent dynamics. Angle-integrated photoelectron spectra $S(E, \Delta t)$ were modeled using multiple exponentially decaying functions, each originating from zero pump–probe delay (Δ*t* = 0) and convolved with the 11-fs experimental cross-correlation $g(\Delta t)$:

$$S(E, \Delta t) = \sum_{i=1,2\ldots} A_i(E) \exp\left(-\frac{\Delta t}{\tau_i}\right) \bigotimes g(\Delta t) \qquad (1)$$

The global fit returns the 1/$e$ decay lifetimes $\tau_i$ and the corresponding energy-dependent amplitudes $A_i(E)$, providing a decay-associated spectrum (DAS) attributable to a dynamical process operating on a specific timescale[15]. For the case of the morpholine data under consideration here, just two exponentially decaying functions were required to produce a satisfactory fit to the experimental measurement. These were found to have decay constants of $\tau_1 = 7 \pm 2$ fs and $\tau_2 = 380 \pm 10$ fs – with the former being extremely close to (and within uncertainty overlap) of $g(\Delta t)$. The DAS extracted from the fitting procedure are shown in Fig. 3, as well as the overall transient signal obtained upon integrating over the low energy (<0.7 eV) region of the time-resolved photoelectron spectrum.

The evolution of the angular anisotropy present in the PAD data was retrieved as a function of photoelectron kinetic energy for each pump–probe timestep using a partial-wave expansion appropriate for describing multiphoton ionization with the linear optical polarization lying parallel to the imaging plane[17,29]:

$$I(E, \Delta t, \theta) = \frac{\sigma(E, \Delta t)}{4\pi}\left[1 + \sum_{L_{even}}^{L_{max}} \beta_L(E, \Delta t)P_L(\cos\theta)\right] \qquad (2)$$

where $\sigma(E, \Delta t)$ is the time-dependent electron energy distribution, the $\beta_L$ terms are the anisotropy parameters describing the shape of the observed PAD, and $P_L(\cos\theta)$ are the $L$th-degree Legendre polynomials. The angles $\theta = 0°$ and 180° lie along the pump and probe laser polarization axes. As already revealed in the time-resolved photoelectron spectrum presented in Fig. 2, ionization proceeds predominantly via a 1+3′ mechanism, with much smaller contributions from 1+4′ processes. Given this observation, the expansion in Eq. 2 was truncated at $L_{max} = 8$, as appropriate for a 4-photon overall interaction. Including terms up to $L_{max} = 10$ gave a negligible change to the results. The PAD anisotropy is related to the angular momentum of the excited state being ionized and is an observable indicative of the electronic character (and its dynamical evolution)[30–34]. The morpholine PAD data displays significant time-dependent variation in all four anisotropy

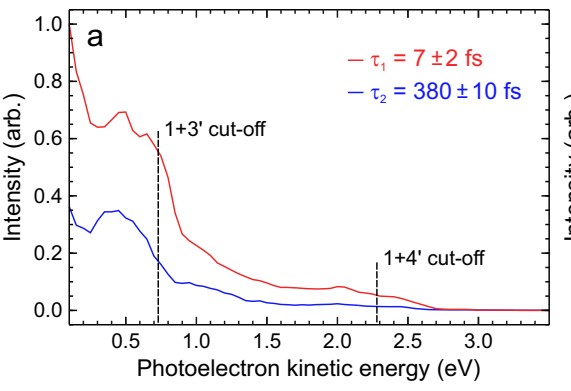

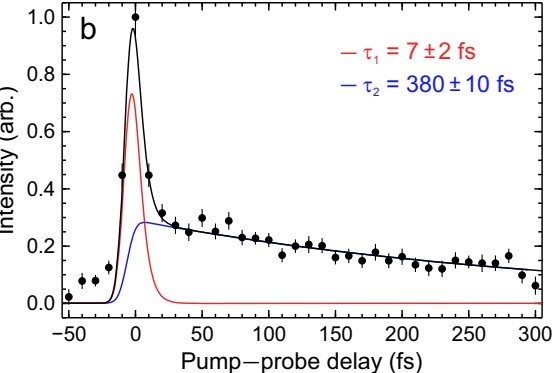

**Fig. 3 | Summary of transient analysis. a** Decay-associated spectra (DAS) plots obtained from the data presented in Fig. 2 using the global exponential fitting approach described by Eq. 1. Uncertainties quoted in the $1/e$ lifetimes are $1\sigma$ values, and the data are partitioned into 0.05 eV energy bins. **b** Transient photoelectron signal integrated over the low kinetic energy region (<0.7 eV), where $1+3'$ ionization is dominant. Error bars are $1\sigma$ values. Solid lines show the corresponding overall fit (black) and contributions from the individual exponential functions (blue and red).

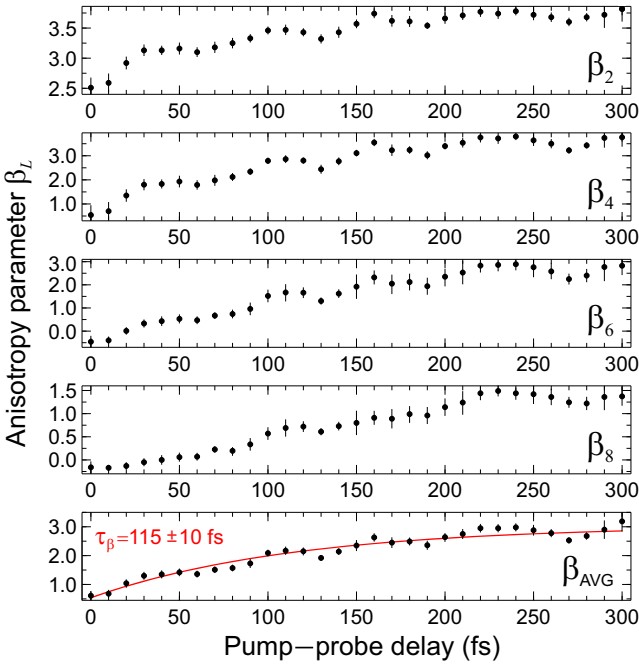

**Fig. 4 | Temporal evolution of the $\beta_L$ anisotropy parameters as a function of pump–probe delay.** Values shown are averages at each timestep over the low photoelectron kinetic energy region (<0.7 eV). Error bars are $1\sigma$ values. The lowest panel is an average across all four sets of $\beta_L$ parameters and includes a single-exponential fit to this data (with a floating amplitude and $y$-axis offset) as a solid red line overlay.

parameters, $\beta_2$, $\beta_4$, $\beta_6$, and $\beta_8$. Figure 4 tracks this behavior, with each of the relevant $\beta_L$ parameters presented as an average over the broad region of low photoelectron kinetic energies, where longer-lived ($\tau_2 = 380 \pm 10$ fs) signals are clearly observed below the $1+3'$ energy cut-off (see Fig. 2 & 3). In all cases, trends in the temporal evolution of $\beta_L$ are consistent over this energy range.

Strikingly, the various anisotropy parameters evolve at a different rate to the dynamical timescales seen in the photoelectron transient signal (Fig. 3), transitioning from their initial values at $\Delta t = 0$ toward a limiting value in under 200 fs. There is, however, some variation here, with the lower $L$ terms converging to a final limiting value slightly more quickly than those associated with higher $L$ values. A single-exponential fit to the average of these data (bottom panel of Fig. 4) reveals a $1/e$ lifetime $\tau_\beta$ of $115 \pm 10$ fs. This is clearly distinct from the

lifetimes revealed by the global fit to the transient photoelectron signal (7 fs and 380 fs) and suggests that the electronic character of the S$_1$(3s/n$\sigma^*$) state may evolve differently from that of the population decay. This evolution is illustrated more dramatically upon reconstructing the PAD at different delay times using the $\beta_L$ anisotropy parameter data from Fig. 4 – as shown in Fig. 5. The PAD undergoes a clear shape change over the first 100 fs or so, before stabilizing after this point.

## Discussion

It can be seen from the transient analysis presented in Fig. 3 that two exponential decay pathways play a significant role in describing the excited-state dynamics of the morpholine molecule following 250 nm absorption. Oliver et al. have previously reported that direct dissociation on the S$_1$(3s/n$\sigma^*$) potential energy surface to form H atom + excited morpholinyl Ã state radicals can only take place at excitation wavelengths <220 nm[19]. This pathway can therefore be ruled out in the present TRPEI measurements, even given the large (~20 nm FWHM) bandwidth of the 250 nm-centered DUV pulses generated using the RDW approach. At wavelengths >220 nm, Oliver et al. suggest that N–H bond fission instead occurs through a CI connecting the S$_1$(3s/n$\sigma^*$) excited-state surface to the S$_0$ ground state, ultimately yielding ground (X̃) state morpholinyl radicals + H atom photoproducts. The extremely fast time constant extracted from our data ($\tau_1 = 7 \pm 2$ fs) is therefore attributed to direct, ballistic passage through this CI via extension of the N–H bond, confirmed by our simulations as described below. As discussed shortly, however, this likely represents a lower bound for this process.

The second time constant seen in our transient data ($\tau_2 = 380 \pm 10$ fs) may be assigned to the more "frustrated" pathway postulated by Oliver et al., in which excited morpholine molecules remain trapped on the S$_1$(3s/n$\sigma^*$) excited-state potential energy surface for a more extended period. The equilibrium structure of ground-state morpholine is pyramidal about the N atom center, and this geometry is initially preserved upon making a vertical (i.e., Franck–Condon) optical transition. In the explanation proposed by Oliver et al., the system evolves toward H atom + morpholinyl Ã state products following photoexcitation but does not pass through the CI connecting the S$_1$(3s/n$\sigma^*$) and S$_0$ states unless the molecule acquires planarity about the amine moiety as the N–H bond extends. This prevents a fraction of the molecules from efficient decay via the CI and leads to a more gradual loss of S$_1$(3s/n$\sigma^*$) state population, as reflected in the slower time constant seen in our data ($\tau_2 = 380 \pm 10$ fs).

To further investigate the above interpretation, the photoexcited dynamics were explored using trajectory surface-hopping simulations

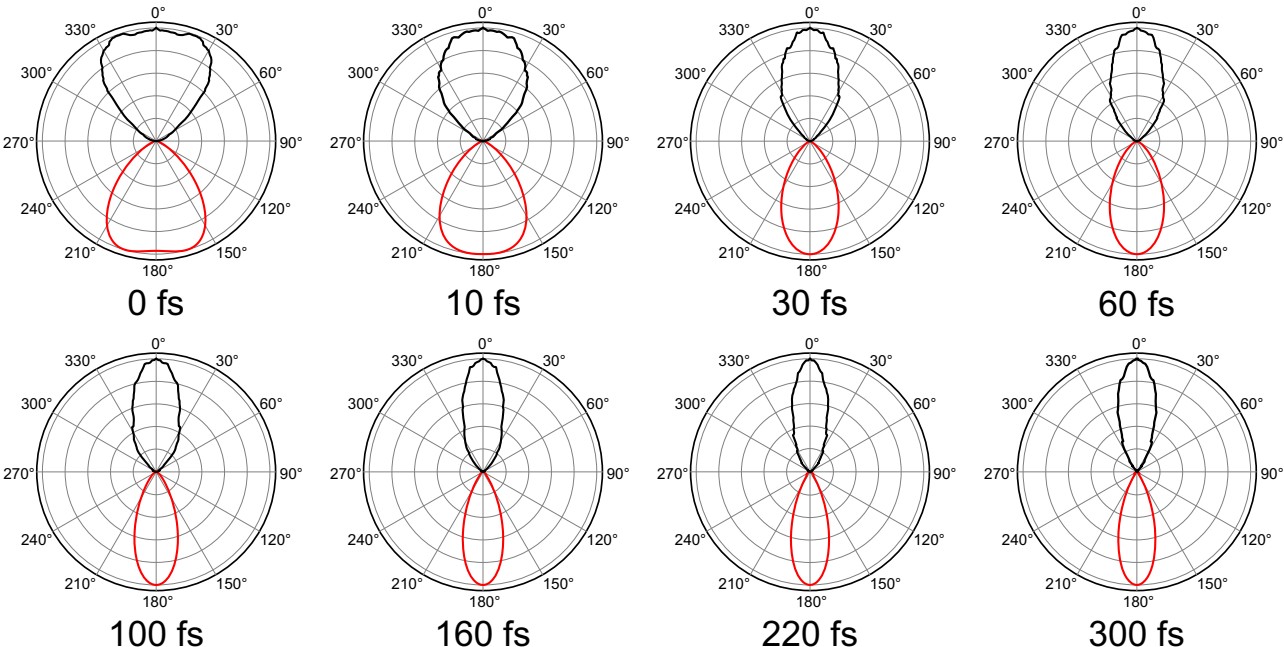

**Fig. 5 | Temporal evolution of PADs.** Normalized polar plots showing the average PAD structure over the low (<0.7 eV) photoelectron kinetic energy region at selected pump–probe delays. The top half of each plot (black line) is the (symmetrized) raw data, while the bottom half of each plot (red line) is generated using Eq. 2 with the $\beta_L$ anisotropy parameters taken from Fig. 4.

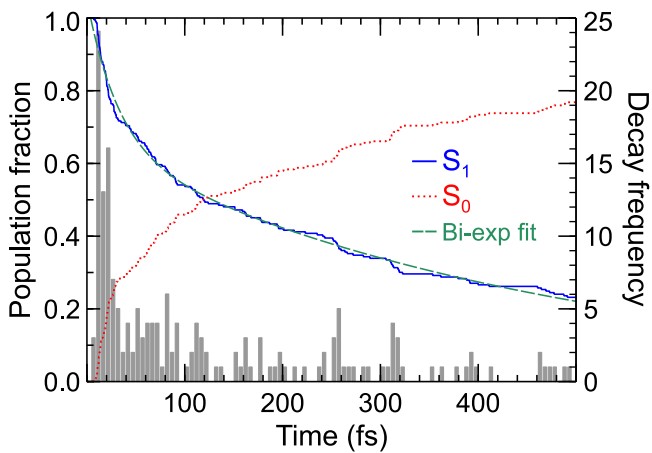

**Fig. 6 | Classical adiabatic populations for the ensemble of surface-hopping trajectories.** The first excited state (blue, solid line) and ground state (red, dotted line) populations are shown for the duration of the simulations (500 fs) and are plotted with respect to the left-hand $y$-axis. The time at which trajectories 'hop' to the ground state is further illustrated by the histogram (gray bars), where each bin spans 5 fs and is plotted with respect to the right-hand $y$-axis. The green dashed line is a bi-exponential fit to the $S_1(3s/n\sigma^*)$ population, yielding time-constants of $\tau_1 = 30 \pm 2$ fs and $\tau_2 = 465 \pm 5$ fs.

using XMS-CASPT2(2e,2o)/aug-cc-pVDZ for the electronic structure (see "Methods" section). All 233 trajectories that were run start in the $S_1(3s/n\sigma^*)$ state, with 77% of these decaying to $S_0$ within 500 fs and the other 23% remaining on the $S_1(3s/n\sigma^*)$ potential surface throughout the simulations. The calculated classical adiabatic populations, shown in Fig. 6, reveal an evolution that correlates with two distinct photophysical pathways operating on two different timescales. The fastest of these is a direct decay pathway in which 18% of the trajectories undergo internal conversion to $S_0$ in under 20 fs. In contrast, 59% decay on a significantly slower timescale, mapping onto the frustrated channel. By fitting the $S_1(3s/n\sigma^*)$ population in a similar manner to the experimental data, we obtain bi-exponential behavior in the

nonadiabatic dynamics with time-constants of $\tau_1 = 30 \pm 2$ fs and $\tau_2 = 465 \pm 5$ fs, in reasonable agreement with the lifetimes extracted from the TRPEI measurement. We note that the extension of the average N–H bond distance within the first 20–30 fs likely leads to a significant reduction in the photoionization cross-section as the $S_1$ state changes from 3s to $n\sigma^*$ character (as expanded upon below)[23,31,35]. This may effectively lead to a slight shortening of the experimentally observed $\tau_1$ lifetime due to windowing. Furthermore, we note that the nuclear dynamics is treated classically in the simulations, thereby excluding nuclear quantum effects. Given the presence of a low barrier along the pathway from the $S_1(3s/n\sigma^*)$ minimum to the $S_1/S_0$ minimum energy conical intersection (MECI, see "Methods" section), quantum tunneling could play a role in this system and contribute to the slightly shorter excited-state lifetimes in the experiment compared to the simulations.

Moreover, our calculations unveil the structural dynamics driving these two distinct photochemical pathways by monitoring the two key internal coordinates that govern the excited-state dynamics in morpholine. These coordinates are identified as the N–H bond length and the pyramidalization angle formed by the intersection of the CNC plane with the N–H bond. We have selected a representative subset of trajectories to demonstrate the characteristic differences in the dynamics for the two channels, as shown in Fig. 7. First, there are trajectories that decay to the ground state in under 20 fs via the $S_1/S_0$ MECI, shown in panel **a**, that correspond to the fast decay pathway responsible for $\tau_1$. Second, illustrative trajectories that decay more slowly, corresponding to the slow frustrated pathway reflected by $\tau_2$, are displayed in panel **b**. In panel **a**, the trajectories start around the $S_0$ minimum (55.12°, 1.02 Å) and evolve directly toward the $S_1/S_0$ MECI (−4.81°, 1.84 Å), bypassing the $S_1(3s/n\sigma^*)$ minimum (−24.94°, 1.07 Å). In contrast, panel **b** shows trajectories that instead evolve toward the $S_1(3s/n\sigma^*)$ minimum, and then continue to oscillate in the pyramidalization angle while being trapped behind the low potential energy barrier separating them from the CI, with average N–H bond distances at around 1.1 Å. Eventually, these trajectories cross the barrier on the $S_1(3s/n\sigma^*)$ surface to access the CI seam with the ground state, leading to dissociation and decay to the $S_0$ state. The different

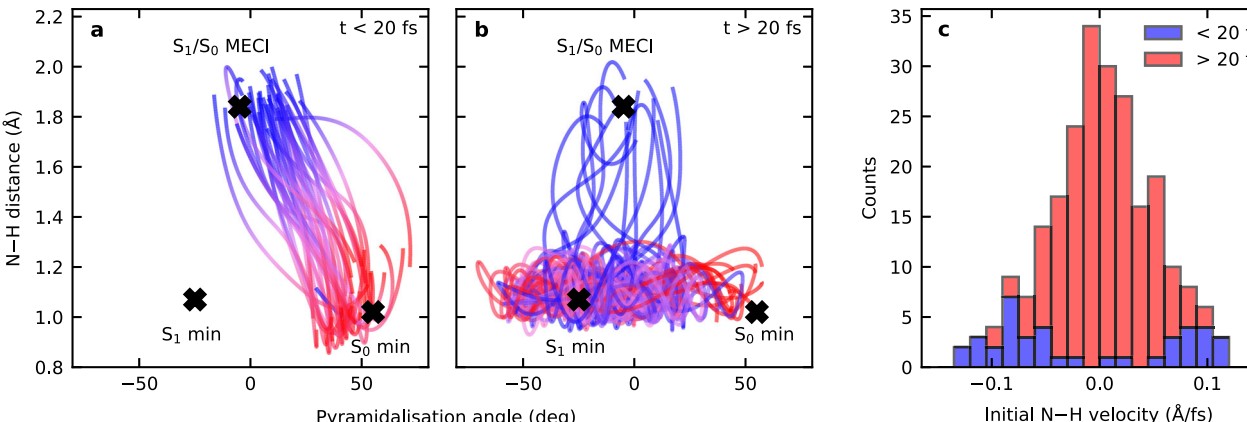

**Fig. 7 | Evolution of N–H bond distance vs. the pyramidalization angle for the fast and the slow channels. a, b** Show traces for exemplary trajectories. The positions of the $S_0$ minimum, $S_1(3s/n\sigma^*)$ minimum, and $S_1/S_0$ MECI are also indicated. **c** Shows the distribution of initial velocities along the N–H bond distance for trajectories corresponding to the slow (red) and the fast (blue) channels.

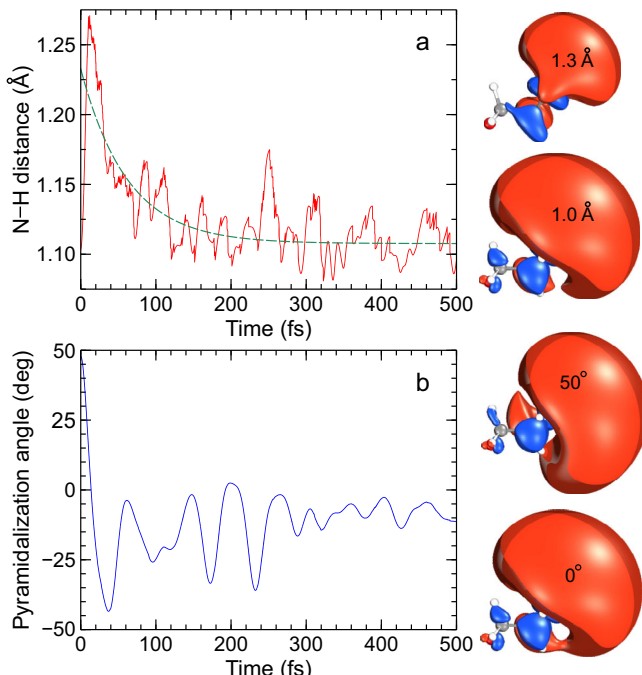

**Fig. 8 | Temporal evolution of selected morpholine internal coordinates. a** Time dependence of the N–H distance for the ensemble of trajectories on the $S_1(3s/n\sigma^*)$ potential energy surface. The $S_1(3s/n\sigma^*)$ molecular orbitals (MOs) have a near-planar pyramidalization angle and show two selected N–H distances. **b** Time dependence of the pyramidalization coordinate. The MOs are now shown at a fixed N–H bond length for two different angular positions. The green dashed line in the upper panel shows a single-exponential fit to the data with a floating $y$-axis offset, and a time constant of $\tau_{NH} = 63 \pm 6$ fs.

dynamics on display in these two sets of trajectories highlight the distinctions between the fast ($\tau_1$) and the slow channel ($\tau_2$). Furthermore, the fast decay is associated with the portion of the initially excited wavepacket that has sufficient momentum along the N–H bond to cross the potential barrier that shields the $S_1/S_0$ CI. This can be seen in panel **c** of Fig. 7, where the fast channel trajectories have larger initial velocity along the N–H coordinate. In contrast, the crossing of the barrier is more statistical in the slower frustrated channel, with comparatively weak correlation between the planarization angle and the propensity of the trajectories to cross the barrier, as seen in panel **b**.

To link the calculations more closely to the experiment, the structural evolution of these two key internal coordinates for the ensemble of trajectories on the $S_1(3s/n\sigma^*)$ potential energy surface is shown in Fig. 8. The average N–H bond distance for the ensemble of trajectories is shown in panel a where an initial spike in this distance is observed at short timescales due to the large number of trajectories which dissociate in the fast regime (attributable to the decay pathway assigned to time constant $\tau_1$). This distance then reduces again on a timescale of 100 fs or so before oscillating rapidly around 1.1 Å. The extension of the N–H bond also induces a significant contraction in the size of the $3s/n\sigma^*$ molecular orbital, transitioning from predominantly diffuse 3s Rydberg character to a much more localized $\sigma^*$ orbital. For the pyramidalization angle, a sharp decrease is observed at short timescales, essentially switching between the equatorial and axial conformations of morpholine. After this, the pyramidalization angle increases again before oscillating around the $S_1(3s/n\sigma^*)$ minimum. As the pyramidalization angle varies (with other coordinates held fixed at the minimum energy geometry), the diffuse nature of the $S_1(3s/n\sigma^*)$ state is retained and moves with the relative orientation of the N–H bond. These properties on the excited state map well onto the observables measured in the experiment, which are mainly sensitive to the $S_1(3s/n\sigma^*)$ population and dynamics.

The data presented in Fig. 4 and Fig. 5 clearly reveal changes in PAD structure evolving on an intermediate timescale that sits in between the numerical $\tau_1$ and $\tau_2$ values that were extracted from transient photoelectron signals (and attributed to the excited-state population dynamics). Since the PAD reflects the electronic character of the state being ionized, this suggests a clear change in this property on a timescale of order 100 fs, with the obvious driver being the evolution of the average N–H bond length in the $S_1(3s/n\sigma^*)$ state. This assertion is reinforced by a single-exponential fit to the average N–H bond length data presented in Fig. 8, which yields a time constant $\tau_{NH} = 63 \pm 6$ fs. This is in good qualitative agreement with the average $\tau_\beta$ of $115 \pm 10$ fs obtained from the temporal analysis of the angular experimental data. Furthermore, we note that the weak modulation in the various anisotropy parameters with time, as seen in Fig. 4, exhibits some resemblance to the oscillatory behavior seen in the pyramidalization angle plot of Fig. 8. We therefore tentatively suggest that periodic evolution of the morpholine geometry in this coordinate may be responsible for these more subtle changes in overall PAD structure. More detailed Fourier analysis of this aspect of the data is, however, inconclusive – although an expanded theoretical exploration of PAD anisotropy and the associated temporal evolution will form the basis of future investigations. Nevertheless, the excellent temporal resolution afforded by RDW-generated pulses in the DUV spectral region –

coupled with the highly differential energy- and angle-resolved information present in our TRPEI measurements – brings detailed new insights to the excited-state dynamics of amine species. In this regard, we note that for the structurally related piperidine molecule (c-$C_5H_{11}N$), Klein et al. similarly invoked a picture of direct vs. frustrated dynamics to rationalize two distinct population lifetimes seen in a TRPEI study using 200 nm excitation[36]. However, that work was conducted with a cross-correlation of $160 \pm 20$ fs, which is over an order of magnitude larger than that used in the present study. Therefore, in that study, no clear distinction between the temporal evolution of the short-time population dynamics and the PAD structure could be resolved.

In summary, the excited-state excess energy redistribution dynamics operating in the morpholine molecule was investigated with extremely high temporal resolution using pump and probe pulses with central wavelengths of 250 nm and 800 nm, respectively. The 250 nm pulse was generated via RDW emission inside a helium-filled HCF, with the direct vacuum-integration of this output leading to a $(1 + 3')$ pump–probe instrument response function of just $11 \pm 2$ fs. Using the TRPEI technique, population dynamics associated with two different dissociation channels originating from the $S_1(3s/n\sigma^*)$ excited state were resolved. The fast channel, exhibiting a time constant of just $7 \pm 2$ fs, was attributed to a ballistic H-atom dissociation through nonadiabatic crossing onto the $S_0$ electronic ground state – although our supporting multireference electronic structure and surface-hopping molecular dynamics calculations indicate this is likely to be a lower bound. This arises due to an experimental windowing effect induced by rapid extension of the average N−H bond distance, and an associated reduction in photoionization cross-section as the $S_1$ state transitions from 3s to $n\sigma^*$ character. The slower channel, with a time constant of $380 \pm 10$ fs, was assigned to a frustrated internal conversion process. This process traps the excited morpholine molecule on the $S_1(3s/n\sigma^*)$ potential energy surface until the potential barrier shielding the CI to $S_0$ can be crossed in an almost statistical manner. Our experimental and theoretical findings broadly confirm the direct and frustrated mechanisms originally proposed by Oliver et al. and reveal the key role of the potential barrier on the $S_1(3s/n\sigma^*)$ state in controlling decay via the CI. Compared to the earlier hypothesis, the correlation between the planarization angle about the nitrogen heteroatom and internal conversion is weaker. Furthermore, the PAD data afforded by our TRPEI measurement reveals that the structural evolution of the morpholine molecule occurs on a distinct timescale to the population dynamics on the $S_1(3s/n\sigma^*)$ state, with an approximate timescale of 100 fs that links to the N−H bond distance.

Our overall results provide new insight into the fundamental photophysics and photochemistry of the N−H chemical bond and clearly highlight the value of obtaining highly differential energy- and angle-resolved data when undertaking time-resolved spectroscopic measurements. More importantly, however, our investigation showcases the potential of optical sources based on the principle of RDW emission to advance the current state-of-the-art in ultrafast science. This is not just through the exceptional time resolution such sources offer, as demonstrated here, but also the potential to exploit broadly tuneable pulses down into the VUV region in a compact tabletop setup. We therefore anticipate that our work will provide an important stimulus for the wider use and further development of such novel sources for a range of spectroscopic applications spanning multiple scientific disciplines.

## Methods
### Experimental
The TRPEI measurements were undertaken using a VMI spectrometer and RDW generation methodology described in detail elsewhere[6,13]. The optical setup produces 250/800 nm pump/probe pulses using a two-stage HCF scheme. In brief, the 800 nm fundamental output of a 1 kHz Ti:Sapphire regenerative amplifier (Spectra-Physics, Spitfire Pro, ~1.5 mJ pulse energy, 55 fs FWHM) was coupled into a first-stage HCF (320 μm core diameter, 1.2 m long) filled with helium at a static pressure of 5 bar. The interaction of the input 800 nm pulse with the helium fill gas induces self-phase modulation and associated spectral broadening (see Fig. 1). The output of this first-stage HCF was then rephased using chirped mirrors and a pair of fused silica wedges, achieving pulse compression down to 10 fs (~0.4 mJ). The compressed beam was then divided into two components using a thin beamsplitter (80 R:20 T, Thorlabs, UFBS8020). The more intense portion of the beam was used for the generation of the DUV pump, and the smaller fraction (further attenuated to ~15 μJ using a neutral density filter) provided the multiphoton probe for the TRPEI measurements. The optical path length in the probe beamline was varied by a computer-controlled translation stage and was incrementally adjusted in 10-fs steps to repeatedly sample pump–probe delay positions between −50 fs and +300 fs. The pump beamline was also attenuated with a neutral density filter (to ~100 μJ) and then coupled into a second-stage HCF (150 μm core diameter, 0.75 m long) using a concave mirror ($f = 50$ cm) to drive the generation of the DUV resonant dispersive wave. These parameters were guided by numerical pulse propagation simulations of the RDW generation process (https://github.com/LupoLab/Luna; https://doi.org/10.5281/zenodo.5513570)[1] and the requirement that the input energy used with this HCF must remain relatively low and be efficiently coupled (>50% transmission) to avoid damage to the tip. The second-stage HCF was integrated directly into the vacuum assembly by gluing its open output end into the ¼-inch protective steel tubing it was housed in. It was then filled with helium (7 bar) at the input end, creating a gradient pressure along the HCF length. This removes the need for a dispersive exit window and leads to extremely short DUV pulse durations at the HCF output (the spectrum centered at 250 nm in Fig. 1 has a FWHM bandwidth of 20.5 nm, corresponding to a Gaussian pulse duration of 4.5 fs FWHM at the Fourier transform limit). Using fresh UV-curing glue (Thorlabs, NOA61) ensures the HCF does not lose tension over time. Previously, when working with argon as the fill gas for the second-stage HCF (<500 mbar input pressure), three stages of differential pumping were required to connect the initial HCF-to-vacuum integration point ($10^{-3}$ mbar) to the VMI region of the photoelectron spectrometer ($10^{-7}$ mbar). The use here, however, of a helium fill at much higher input pressures necessitates the addition of a fourth differential pumping stage. This took the form of a small cell that was positioned directly at the output end of the HCF and maintained at $10^{-1}$ mbar using a high-throughput rotary pump (Edwards E2M40). The cell was then coupled to a larger vacuum chamber where optical manipulation of the RDW beam could take place – as used in our earlier setup with an argon fill gas[6] – via a narrow orifice channel (1.5 mm diameter, 40 mm long).

The DUV RDW output was recollimated under vacuum to ~2 mm in diameter using a concave aluminum mirror ($f = 50$ cm) and then separated from the residual co-propagating IR driving pulse using a pair of dichroic mirrors (Edmund Optics, 47–985). The input into the RDW HCF was attenuated such that the DUV pulse output energy after the dichroic high reflectors was ~3 μJ. It was unnecessary to generate higher pulse energies (up to 10 μJ could be achieved) as it is desirable for the experimental conditions to remain in the weak-field (i.e., multiphoton ionization) regime and to also preserve the usable lifetime of the second-stage HCF (~80 h). The RDW output was wavelength-tunable as a function of helium pressure, but a central wavelength of 250 nm was selected here to match the $S_1(3s/n\sigma^*)$ absorption band in the morpholine molecule. By reflecting the DUV pump pulse from an aluminum-coated D-shaped mirror and propagating the IR probe past the flat edge of this optic, the two beamlines were made to propagate co-linearly. The pump and probe then entered the main vacuum chamber of the TRPEI spectrometer, initially passing unfocused through a set of electrostatic VMI lenses before reflecting off a curved

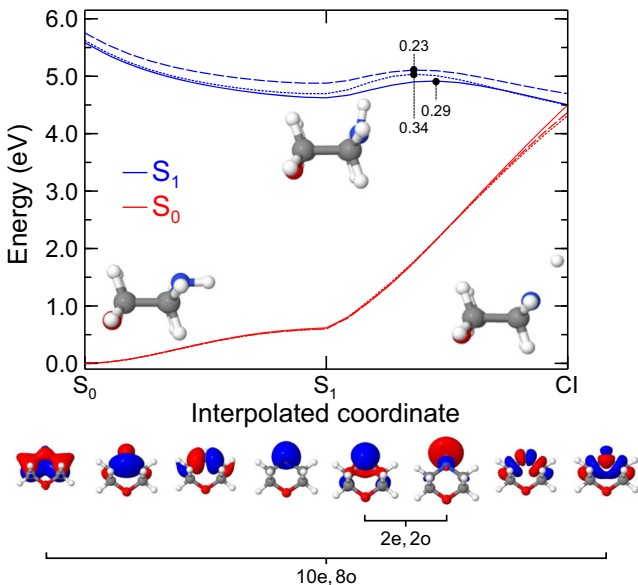

**Fig. 9 | Ground (S₀) and excited-state (S₁) energies along the LIIC geometries from the S₀ minimum to the S₁ minimum, and then to the MECI.** The solid and dotted lines represent output obtained using the XMS-CASPT2(2e,2o) and (10e,8o) active spaces, respectively, while the dashed line plots results from CC3. The orbitals in the two active spaces are shown visually under the main plot. All geometries were computed at the XMS-CASPT2(2e,2o) level. Small variations in the barrier height to the CI could potentially affect the time constant for the slow photodissociation channel.

**Table 1 | Geometrical parameters of various critical points**

| Critical point | $R_{N-H}$ (Å) | $\theta$ (deg.) |
|---|---|---|
| S₀-min | 1.02 | 55.12 |
| S₁-min | 1.07 | −24.94 |
| S₀–S₁ CI | 1.84 | −4.81 |

Geometries were computed at the XMS-CASPT2(2e,2o) level of theory.

aluminum mirror ($f = 10\,cm$) mounted on a high-precision $x$–$y$–$z$ manipulator. The tightly focused and spatially overlapped optical pulses then ionized a skimmed molecular sample beam on a second optical pass back through the VMI region. This sample beam was generated by passing helium carrier gas (0.5 bar) through a room temperature reservoir of morpholine and then allowing a continuous flow expansion into vacuum via a 150 μm pinhole. Photoelectrons produced during the ionization event were imaged using a 40 mm diameter dual micro-channel plate/P47 phosphor screen detector in conjunction with a CCD camera (640 × 480 pixels). Focused optical intensities at the interaction region were of order $10^{13}\,W\,cm^{-2}$ for both the pump and the probe, yielding associated Keldysh parameters[37] >1. This suggests that strong-field tunnel ionization effects are not expected to be significant – as confirmed in our experimental observations (Figs. 2 and 3). Although the Keldysh parameter is known to be a limited approximation for molecular ionization[38], it still serves as a useful indicative benchmark when evaluating our starting experimental conditions.

**Theoretical**
Extended multi-state complete active space with second-order perturbation theory (XMS-CASPT2) calculations were performed using the BAGEL 1.1 software package (BAGEL: Brilliantly Advanced General Electronic-structure Library. http://www.nubakery.org under the GNU General Public License.)[39]. Coupled cluster singles and doubles with

perturbative triples method (CC3) calculations were performed with $e^{T}$ [40]. Trajectory surface-hopping simulations were undertaken using the SHARC package (Version 2.1.2; https://github.com/sharc-md/sharc/releases/tag/v2.1.2) utilizing BAGEL for the electronic calculations[41]. In all XMS-CASPT2 calculations, the active space was state-averaged across the ground and lowest-energy singlet excited state. Additionally, a real shift of 0.3 au was employed, a single-state single-reference treatment was used for the multi-state component, and all 1s-core-orbitals on non-hydrogen atoms were frozen in the orbital optimization procedure. Unless stated otherwise, all calculations use the aug-cc-pVDZ basis set, with density fitting.

**Electronic-structure benchmarking**
Prior to undertaking the dynamics simulations, a benchmarking study of the electronic structure was undertaken. The smaller active space considered, XMS-CASPT2(2e,2o), consists solely of the nitrogen lone pair and the mixed 3s Rydberg/σ* orbitals, which, upon extension of the N−H bond, localizes to a σ* anti-bonding orbital. A larger XMS-CASPT2(10e,8o) approach includes the addition of the N−H σ orbital into the active space, in addition to encompassing σ and σ* orbitals across the CNC bridge (see Fig. 9 for further information). Three critical points were determined at the XMS-CASPT2(2e,2o) level of theory: the equatorial ground state (S₀) minimum; the lowest-energy singlet state S₁(3s/nσ*) minimum; and the minimum energy conical intersection (MECI) between S₀ and S₁(3s/nσ*), all of which are summarized in Table 1 and Fig. 9. The S₀ and S₁(3s/nσ*) geometries were validated as minimum energy structures via numerical Hessian calculations. The S₀ equatorial arrangement was found to be 0.06 eV lower in energy than the S₀ axial conformer. Under the cold molecular beam expansion conditions of our experiment (assumed ≤100 K), Boltzmann statistics indicate a negligible population of <1% for the axial conformer, and so this was not considered further. A hypothetical reaction pathway was constructed by performing a linear interpolation in internal coordinates (LIIC) from S₀ to the S₁(3s/nσ*) minimum, and then onward to the S₁/S₀ MECI. In terms of the S₀–S₁(3s/nσ*) pathway, the XMS-CASPT2(2e,2o) and XMS-CASPT2(10e,8o) surfaces are very similar when compared with the CC3 calculations – which serve to benchmark our active space methods against a black-box, albeit single-reference, approach. Along this pathway, the CC3 surface is slightly higher in energy but essentially remains parallel to the multireference results. The main structural change along the LIIC involves an evolution in the pyramidalization angle while the N−H distance remains constant. For the S₁(3s/nσ*) to MECI part of the LIIC, the main structural changes involve a combination of planarization about the N atom center and an extension of the N−H bond. On the XMS-CASPT2(2e,2o) potential energy surface, a barrier on the S₁ state of 0.29 eV is encountered, which is comparable to the barrier heights for both the larger active space (0.34 eV) and the CC3 (0.23 eV) calculations. We note that these barrier heights are likely an upper bound to the true value, as these appear along the LIIC. Furthermore, our electronic-structure benchmarking is only carried out up to the MECI since the experiment only measures the population of the S₁(3s/nσ*) state. In conclusion, these results indicate that XMS-CASPT2(2e,2o) is sufficiently robust and accurate for the nonadiabatic dynamics simulations.

**Nonadiabatic dynamics**
To take into account anharmonic and zero-point energy effects[42–44], initial conditions were sampled from MP2/cc-pVDZ Born-Oppenheimer molecular dynamics with a quantum thermostat (BO-QT) implemented in ABIN[45–47]. The thermostat parameters were taken from the GLEMD website (http://gle4md.org/), employing $Ns = 6$, $\hbar\omega/kT = 20$, and strong coupling, with a target temperature of 298 K. A comparatively high value of temperature is used to account for zero-point energy in the system. After an equilibration period, 9991 uncorrelated geometries were sampled. From these, initial conditions

were sampled in the range of 4.90–5.30 eV. In the dynamics, a total of 233 nonadiabatic trajectories were propagated using XMS-CASP-T2(2e,2o) and trajectory surface-hopping for 500 fs or until a hop to the electronic ground state occurred. Trajectories were propagated using classical timesteps of 0.1 fs in the MCH representation with 25 sub-steps using a local diabatization scheme (wavefunction overlaps).

## Data availability

Data relating to this study are available at https://doi.org/10.17861/90cd261f-1b7a-40df-a165-ec2ac8f3fbaa.

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

## Acknowledgements

This work was primarily supported by Engineering and Physical Sciences Research Council (EPSRC) grants EP/P001459/1 (D.T., M.J.P.), EP/R030448/1 (D.T., J.C.T., M.J.P., N.K.), EP/T021675/1 (M.J.P.), and EP/V006746/1 (M.J.P., D.T., A.K.), as well as the European Research Council under the European Union's Horizon 2020 Research and Innovation program: Starting Grant agreement HISOL, No. 679649 and Consolidator Grant agreement XSOL, no. 101001534 (J.C.T.). Further support was also provided through EPSRC grants EP/T020903/1 (J.C.T.), EP/V006819/2 (A.K.), EP/V049240/2 (A.K.), EP/X026698/1 (A.K.), EP/X026973/1 (A.K.), and the U.S. Department of Energy, Office of Science, Basic Energy Sciences, under award number DE-SC0020276 (A.K.). A.W.P., L.H., A.K., and M.J.P. gratefully acknowledge the Leverhulme Trust through research project grant RPG-2020-208. J.C.T. is supported by a Chair in Emerging Technology from the Royal Academy of Engineering. C.B. acknowledges support from the Royal Academy of Engineering through Research Fellowship No. RF/202122/21/133. PhD funding for S.L.J. and C.S. from Heriot-Watt University and for L.B. from the University of Oxford is also acknowledged.

## Author contributions

D. Townsend coordinated the overall project with supporting contributions from J. C. Travers (experiment) and M. J. Paterson/A. Kirrander (theory). S. L. Jackson undertook the TRPEI measurements after modifying and upgrading an RDW optical beamline setup originally designed and built by N. Kotsina and C. Brahms. The supporting theory calculations were performed by A. W. Prentice, L. Bertram, and L. Hutton. Analysis of the experimental data was conducted by S. L. Jackson. Codes for pBASEX image processing were developed by C. Sparling. The paper was written by S. L. Jackson and D. Townsend, with additional contributions from C. Brahms, A. W. Prentice, L. Bertram, L. Hutton, A. Kirrander, and M. J. Paterson. All authors contributed to the discussion of the results.

## Competing interests

The authors declare no competing interests.
