## [Peer review file · Nature Communications]

Decoupling structural molecular dynamics from excited state lifetimes using few-femtosecond ultraviolet resonant dispersive waves

Corresponding Author: Professor Dave Townsend

Version 0:

Reviewer comments:

Reviewer #1

(Remarks to the Author)

In this manuscript, the authors investigate excited-state dynamics of gas-phase morpholine using time-resolved photoelectron imaging technique in combination with few-femtosecond ultraviolet resonant dispersive wave generation in a helium-filled hollow capillary fiber. They successfully performed a pump-probe experiment with an exquisite temporal resolution (instrument response function of ~11 fs at FWHM). The experimental data are of good quality. Additionally, high-level multi-reference electronic structure calculations coupled with non-adiabatic surface hopping molecular dynamics were performed to support their interpretation of the experimental findings (the 7 and 380 fs time constants, and the variation of the PADs). The paper is well written. I think that it is publishable at Nature Communications. However, I do have a few concerns:

- 1) I note that the absorption cross-sections for morpholine molecules at ~250 nm are relatively small (M. Al Rashidi et al., Atmospheric Environment 88 (2014) 261-268). I would expect two-color multiphoton non-resonant ionization to also contribute to signals around time zero, which should be hard to separate from the 7 fs component, especially with scan step size of 10 fs (Fig. 3). Comments on this point are suggested.
- 2) In Fig. 4, the fitting PADs are suggested to be shown together with experimental PADs, so the fitting quality can be presented.
- 3) For theoretical calculations about nonadiabatic dynamics of morpholine molecule, the authors state that "initial conditions were sampled in the range of 4.90 to 5.30 eV with a hypsochromic shift of 0.13 eV applied to align with the experimental absorption value ($\lambda_{\max} = 233 \text{ nm}/5.56 \text{ eV}$)". Is "233 nm/5.56 eV" should be "223 nm/5.56 eV" or "233 nm/5.32 eV"? Is the initial wavepacket at 250 nm/4.96 eV or not?

Reviewer #2

(Remarks to the Author)

Reviewer #3

(Remarks to the Author)

Jackson et al. used time-resolved photoelectron imaging (TRPEI) measurements to explore the N-H bond dissociation of morpholine. In their study, they generated the 250 nm deep-UV excitation pulses with a resonant dispersive wave (RDW) generated in a hollow core fiber. The time and angular dependence of the TRPE images allow them to infer the timescales and mechanisms for bond dissociation with high time resolution (see below). These findings are greatly complemented by corresponding high-level ab initio theory and dynamics.

In my view, the major accomplishment of this work is to demonstrate the wider utility of RDW laser technology to produce <

10 fs pulses, and the potential insights it can bring to the Chemical Physics community. While I am, in principle, supportive of the manuscript, there are important (one major technical) issues that must be addressed.

1) Jackson et al. claim the instrument response is 11 ± 2 fs (see Figure 1). This may be true, but the data is not sufficiently sampled in the time domain to justify this conclusion. If properly sampled, it may reveal side-bands or unwanted structure, indicating a longer (and perhaps non-Gaussian) instrument response. For a FWHM of $\sigma = 11$ fs, I would expect to see > 6 points per σ , whereas the authors appear to have used $\sim 3-4$.

2) The low sampling of the pump-probe time delay (especially around time zero) brings into question the reliability of the first time constant (7 ± 2 fs, Figure 3) quoted for the fast N-H bond dissociation. I would also add that given the time constant retrieved from fitting is shorter than the IRF, and that sampling is low, the authors give the IRF as the lower bound for this time constant.

3) The time constant for 'direct' N-H bond dissociation is presented in two disconnected ways: 7 fs from fits to TRPEI PADs, and a secondary more subtle way: (quoting the authors from page 13): "We note that the extension of the average N-H bond distance within the first 20–30 fs likely leads to a significant reduction in the photoionization cross-section as the S1 state changes from $3s$ to $n\sigma^*$ character (as expanded upon below). This may effectively lead to a slight shortening of the experimentally observed τ_1 lifetime due to windowing." With theory then concluding the N-H bond dissociation (prompt) timescale is around 30 fs.

So how can these two differing time constants be reconciled? The abstract and conclusions state that the primary N-H bond dissociation time constant is 7 fs, but this doesn't seem to be the full (and correct) explanation returned from high level theoretical calculations.

More minor comments:

a) There are two conformers of gas-phase morpholine, where the N-H bond can take axial or equatorial conformations. Did the ab initio calculations identify any dynamical differences between the two conformers and how they approach the $3s/n\sigma^*$ conical intersection?

b) The H (Rydberg) atom photofragment translational spectroscopy experiments from the Ashfold group are not 'static spectroscopic' measurements as the authors suggest in multiple instances in the manuscript (pages 2 and 5). Instead, these are ultimately frequency domain measurements capable of measuring the vibrationally resolved product distribution of the organic radical after N-H bond dissociation.

c) Reading Figure 6 and the text surrounding it on page 14, I was confused whether the S0 here is from a direct internal conversion back to the starting state, or instead the authors meant the bond dissociation co-ordinate which is adiabatically linked to S0? I believe the latter.

Reviewer #4

(Remarks to the Author)
See attached PDF.

Reviewer #5

(Remarks to the Author)

Version 1:

Reviewer comments:

Reviewer #1

(Remarks to the Author)

The author has addressed all my questions and concerns in last review report. I have no further questions. I recommend to publish this manuscript as it is.

Reviewer #2

(Remarks to the Author)

Reviewer #3

(Remarks to the Author)

The revisions the authors have made to the manuscript have improved the paper. I have several remaining comments that the authors need to address:

Comment 1:

I appreciate the experimental problems that the authors report with backlash from their stage. However, this does result in a lower than necessary sampling of the pump-probe time delay. To demonstrate the reproducibility of their IRF and qualify their additional statement, they have added to the revised manuscript, they should also show the repeat IRF measurements in the supporting information.

Comments 2:

Some groups indeed choose to quote time constants below their IRF. However, in those cases, the sampling of the pump-probe time delay is higher than in the present study. Therefore, the authors should quote the time constant as IRF limited rather than 7 ± 2 fs, especially as they have carefully reworded the manuscript to indicate this is a lower bound to time constant associated with the prompt first N-H bond dissociation channel.

Comment 4:

There are two points I would like the authors to reconsider in their response: 1) the molecular beam temperature is not characterised, and so it is unclear what the relative Boltzmann populations will be. 2) A combined experimental and theoretical study (ref. 19) determined the energy between the ground state axial and equatorial conformers to be 0.022 eV, three times lower than the calculated value here.

Comment 6:

For clarity, the authors need to define whether they are referring to the adiabatic or diabatic S1 and S0 states. I believe it is adiabatic. If the reader is not in this mindset, it complicates reading the manuscript. Briefly stating this or reminding the reader when necessary will be helpful. Especially to clarify what is meant by "decay to the S0 state", a phrase which is used multiple times in the manuscript.

Reviewer #4

(Remarks to the Author)

The revised manuscript has solved our previous concerns. From our site, we believe that it meets the criteria of scientific research published in Nature Communications.

Reviewer #5

(Remarks to the Author)

Decoupling structural molecular dynamics from excited state lifetimes using few-femtosecond ultraviolet resonant dispersive waves (NCOMMS-25-22152)

Response to Specific Comments:

Reviewer #1&2

Comment 1: *I note that the absorption cross-sections for morpholine molecules at ~250 nm are relatively small (M. Al Rashidi et al., Atmospheric Environment 88 (2014) 261-268). I would expect two-color multiphoton non-resonant ionization to also contribute to signals around time zero, which should be hard to separate from the 7-fs component, especially with scan step size of 10 fs (Fig. 3). Comments on this point are suggested.*

Response: There is certainly a possibility of non-resonant ionization close to zero pump-probe delay. These signals would, however, be expected to vary with the intensity of the 800 nm multiphoton probe in a different way to those originating from any resonant (1+3') process. Upon varying the probe intensity, we see no appreciable change in the relative strength of the signals associated with the short (7 fs) and long (380 fs) dynamics (where the latter can only be from a resonant ionization). This indicates that non-resonant effects are not a significant factor in our measurement.

Changes: This is a helpful point, and we have added to the text on page 7 for clarification: *“Upon varying the 800 nm probe intensity, no appreciable change was observed in the relative strength of the ionization signals obtained close to $\Delta t = 0$ and at longer pump-probe delay times. This indicates that two-colour non-resonant ionization is not a significant factor in our measurements.”*

Comment 2: *In Fig. 4, the fitting PADs are suggested to be shown together with experimental PADs, so the fitting quality can be presented.*

Response: We think this comment in fact refers to Fig. 5. It's a very good point to raise – particularly because in updating this figure it became apparent that we had made an error in plotting the PADs! This doesn't affect any of our overall interpretation or conclusions but it's good it was picked up.

Changes: Fig. 5 has been replotted so that the experimental data is also shown. The accompanying caption has also been updated.

Comment 3: *For theoretical calculations about nonadiabatic dynamics of morpholine molecule, the authors state that “initial conditions were sampled in the range of 4.90 to 5.30 eV with a hypsochromic shift of 0.13 eV applied to align with the experimental absorption value ($\lambda_{max} = 233 \text{ nm}/5.56 \text{ eV}$)”. Is “233 nm/5.56 eV” should be “223 nm/5.56 eV” or “233 nm/5.32 eV”? Is the initial wavepacket at 250 nm/4.96 eV or not?*

Response & Changes: Thank you for bringing this to our attention. It should be $\lambda_{max} = 223 \text{ nm}/5.56 \text{ eV}$. We have corrected the accordingly on page 26.

Reviewer #3

Comment 1: *Jackson et al. claim the instrument response is 11 +/- 2 fs (see Figure 1). This may be true, but the data is not sufficiently sampled in the time domain to justify this conclusion. If properly sampled, it may reveal side-bands or unwanted structure, indicating a longer (and perhaps non-Gaussian)*

instrument response. For a FWHM of $\sigma = 11$ fs, I would expect to see > 6 points per σ , whereas the authors appear to have used ~ 3 -4.

Response: The sampling interval used here is a consequence of backslashing issues associated with our translation stage when taking < 10 fs timesteps across multiple pump-probe delay scans. We are, however, confident that there are no additional side bands or structure in our data. This is based on a consideration of several data sets recorded across multiple days (of which just one is shown in Fig. 1), each of which has a slightly different offset in the true zero pump-probe delay position. In all cases, the fitted IRF is well replicated by a simple Gaussian function.

Changes: The text on page 4 has been extended as follows: “A Gaussian fit to these data yields a FWHM of just 11 ± 1 fs. *This is reproducible day-to-day and is consistent under small experimental shifts in the temporal sampling with respect to the true zero pump-probe delay position*”.

Comment 2: *The low sampling of the pump-probe time delay (especially around time zero) brings into question the reliability of the first time constant (7 ± 2 fs, Figure 3) quoted for the fast N-H bond dissociation. I would also add that given the time constant retrieved from fitting is shorter than the IRF, and that sampling is low, the authors give the IRF as the lower bound for this time constant.*

Response: This is a good point to raise but we note the fitting a single $1/e$ time constant that is shorter than the IRF is generally reasonable (and is frequently reported by many groups) as long as $\tau > \text{FWHM}/2$ and the signal-to-noise ratio is good – as is the case here. Given the next comment raised below about windowing effects, our quoted lifetime of $\tau_1 = 7 \pm 2$ fs is certainly expected to be a lower bound – as we discuss towards the end of the manuscript. This is not, however, a limitation of our sampling, but a more subtle effect arising due to variation in the photoionization cross-section of the $S_1(3s/n\sigma^*)$ state over time as the average molecular geometry evolves.

Changes: We have taken steps to make it clearer throughout the manuscript that the $\tau_1 = 7 \pm 2$ fs represents a lower bound on the lifetime associated with the direct N-H fission pathway.

In the Abstract on page 1: “We observe two distinct pathways leading to N–H bond fission: a *rapid* direct process (*with a lower bound of 7 ± 2 fs*) and a more frustrated mechanism (380 ± 10 fs) with hindered access to the conical intersection connecting with the electronic ground state.”

On page 12: “The extremely fast time constant extracted from our data ($\tau_1 = 7 \pm 2$ fs) is therefore attributed to direct, ballistic passage through this CI via extension of the N–H bond, confirmed by our simulations as described below. *As discussed shortly, however, this likely represents a lower bound for this process.*”

On page 19: “The fast channel, exhibiting a time constant of just 7 ± 2 fs, was attributed to a ballistic H-atom dissociation through non-adiabatic crossing onto the S_0 electronic ground state – *although our supporting multi-reference electronic structure and surface hopping molecular dynamics calculations indicate this is likely to be a lower bound. This arises due to an experimental windowing effect induced by rapid extension of the average N–H bond distance, and an associated reduction in photoionization cross-section as the S_1 state transitions from $3s$ to $n\sigma^*$ character.*”

Comment 3: *The time constant for ‘direct’ N-H bond dissociation is presented in two disconnected ways: 7 fs from fits to TRPEI PADs, and a secondary more subtle way: (quoting the authors from page 13): “We note that the extension of the average N–H bond distance within the first 20–30 fs likely leads to a significant reduction in the photoionization cross-section as the S_1 state changes from $3s$ to $n\sigma^*$ character*

(as expanded upon below). This may effectively lead to a slight shortening of the experimentally observed τ_1 lifetime due to windowing.” With theory then concluding the N–H bond dissociation (prompt) timescale is around 30 fs.

So how can these two differing time constants be reconciled? The abstract and conclusions state that the primary N-H bond dissociation time constant is 7 fs, but this doesn't seem to be the full (and correct) explanation returned from high level theoretical calculations.

Response & Changes: See response to Comment 2 above: the time constant extracted from our experiments is likely to be a lower bound due to the windowing effect. This explains the difference in time constants obtained from experiment and theory

Comment 4: *There are two conformers of gas-phase morpholine, where the N-H bond can take axial or equatorial conformations. Did the ab initio calculations identify any dynamical differences between the two conformers and how they approach the 3s/nsigma* conical intersection?*

Response: This is a good point; however, under the molecular beam conditions of the experiment we only expect a very small fraction (<1%) of the population to be in the axial conformer. This was calculated using a Boltzmann distribution and the energy difference between the optimised geometries of the two conformers (0.06 eV). Therefore, since almost all the population should be in the lower energy equatorial conformer, we did not investigate the axial conformer any further.

Changes: We have modified the text on page 24 to make it clear why we only consider the lower-energy equatorial conformer with: “The S₀ equatorial arrangement was found to be 0.06 eV lower in energy than the S₀ axial conformer. Under the cold molecular beam expansion conditions of our experiment, Boltzmann statistics indicate a negligible population of <1% for the axial conformer, and so this was not considered further.”

Comment 5: *The H (Rydberg) atom photofragment translational spectroscopy experiments from the Ashfold group are not ‘static spectroscopic’ measurements as the authors suggest in multiple instances in the manuscript (pages 2 and 5). Instead, these are ultimately frequency domain measurements capable of measuring the vibrationally resolved product distribution of the organic radical after N-H bond dissociation.*

Response: We appreciate that our phrasing could be easily changed here for improved clarity.

Changes: The manuscript text has been modified as follows:

Page 2: “The excellent time resolution afforded by this approach allows us to identify two distinct pathways governing molecular photofragmentation that have been inferred *previously from frequency domain measurements interrogating the vibrationally resolved distribution of organic radical products* but never observed directly.”

Page 5: “In such a *frequency-resolved* measurement, however, only the asymptotic photoproduct limit is probed, and these pathways cannot be observed directly.”

Comment 6: *Reading Figure 6 and the text surrounding it on page 14, I was confused whether the S₀ here is from a direct internal conversion back to the starting state, or instead the authors meant the bond dissociation co-ordinate which is adiabatically linked to S₀? I believe the latter.*

Response: We thank the reviewer for their helpful comment. All trajectories were initiated on the S_1 state in the simulations. After crossing the barrier on the S_1 state, trajectories can decay to the S_0 state through a conical intersection, which can only be accessed when the N—H bond extends and the pyramidalization angle is planar (shown in Figure 7 and 9).

Changes: To improve the clarity of the text we have inserted the following changes:

Page 15: “First, there are trajectories that decay to the ground state in under 20 fs **via the S_1/S_0 MECI**, shown in panel a, that correspond to the fast decay pathway responsible for τ_1 .”

Page 15: “Eventually, these trajectories cross the barrier on the $S_1(3s/n\sigma^*)$ surface to access the CI seam with the ground state, leading to dissociation **and decay to the S_0 state.**”

Reviewer #4&5

General Opening Comment: *The authors comments that the RDW would be possible to revolutionize ultrafast science over next decade, which is overestimate the fessibility of the RDW phenomenon and associated ultrafast technique. In basic, the RDW associated ultrashort pulse even close to sub-cycle attosecond light field theoretically, its influence could not be exaggerated to revolutionizing ultrafast science, the past one who make it is the high harmonic generation and attosecond pulse generation.*

Response: *A key aspect of the RDW approach is that is provides extremely bright (up to 10s μ J), very short (<5 fs), and fully tuneable optical pulses in the deep ultraviolet (DUV, 300-200 nm). This spectral region is of great importance as it aligns with the lowest-lying electronic absorption bands of many molecules. RDW-based sources therefore present new opportunities to undertake ultrafast spectroscopy/dynamics measurements with the best possible temporal resolution when pumping these absorption bands. While such measurements may be coupled with an XUV probe pulse derived from high-harmonic generation (HHG) – see *Optica* **11**, 1320 (2024) for a recent proof-of-concept demonstration – the critical point is that accessing the DUV region with HHG-based approaches is very challenging and, even when achieved, generates insufficient pulse energy to act as an effective pump. As a result, even some of the fastest experiments using HHG probes report similar instrument response functions to our work. Attosecond pulses in the DUV from an RDW source have just been demonstrated (see *Nat. Photon.* (2025), <https://doi.org/10.1038/s41566-025-01658-5>), which further highlights their huge potential. We therefore stand by our assertion that RDW sources are set to play a revolutionary role in ultrafast science moving forwards.*

Changes: *We have added to the text on page 2 to make the above points clearer: “Because it can efficiently create tuneable VUV/DUV pulses with few-femtosecond duration – **something that is very challenging via other means** – the RDW phenomenon is rapidly emerging as a novel technique for ultrafast spectroscopy which will play a leading role in next-generation ultrafast science. Such time-resolved investigations are relevant to a broad range of sub-disciplines within chemistry, physics, biology and materials science – **where many systems of interest show the onset of electronic absorption bands in the DUV and near VUV spectral regions.** The impact of RDW-based light sources will therefore be far-reaching.”*

Comment 1: *Abstract * As mentioned above, the introduction of RDW and its application in ultrafast phenomenon should be constricted to certain directions and areas. * revise all the sentences with extremely, excellent. Acuturally, the time resolution of the ultrafast dynamics measurement is far below the approach based attosecond pulse through HHG. * Please revise the conclusion of ... is not possible*

without the A VUV pulse generated through high harmonic generation could reach the same effect, by including the additional field locking methods, the time resolution could be even higher than the current pump-probe metrology in this manuscript. So that a fair and humble expression is very necessary. The experimental descriptions have the same issue throughout the whole manuscript, and the words suddenly return to normal status in the theoretical part. * The theoretical demonstration is very important in this manuscript, its role was excessively weakened in the introduction.

Response & Changes: Breaking this down point-by-point:

* The introduction of RDW and its application in ultrafast phenomenon should be constricted to certain directions and areas

We disagree – the potential of RDW-based optical sources will be broadly applicable to a wide range of problems in ultrafast science. Please also see the discussion provided in the response to Comment 2 below.

* Revise all the sentences with extremely, excellent. Accurately, the time resolution of the ultrafast dynamics measurement is far below the approach based attosecond pulse through HHG

We highlight our response to the General Opening Comment above here. We have, however, removed the instance of “extremely short” in the Abstract when referring to the optical pulse duration. This text now reads: “Ultraviolet excitation with a 250 nm pump pulse was achieved via RDW emission inside a helium-filled hollow capillary fibre”. We have also changed “extremely fast” to “rapid” when discussing the direct N–H bond breaking process. We do, however, stand by the phrase “When used in conjunction with a short 800 nm probe, an instrument response function of just 11 ± 1 fs was realized, permitting spectroscopic measurements with excellent temporal resolution”. To the best of our knowledge, this instrument response function is better than anything previously reported by other research groups when using the time-resolved photoelectron imaging technique. It is also comparable to some of the fastest experiments which use attosecond pulses generated with HHG in conjunction with a UV/Vis or infrared pump pulse.

* Please revise the conclusion of ... is not possible without the

We have revised the text in the Abstract to: “*Observing this clean distinction between population lifetimes and structural dynamics would be very challenging without the optical properties afforded by the RDW approach*”.

* The theoretical demonstration is very important in this manuscript, its role was excessively weakened in the introduction.

On reflection, this is a good point to raise. We have amended the Abstract text to better promote the role of theory: “*Data interpretation is supported by high-level multi-reference electronic structure calculations coupled with non-adiabatic surface hopping molecular dynamics, and this synergy between experiment and theory is vital for developing a complete mechanistic picture.*”

Comment 2: Revise this sentence. ... which will play a leading role in next-generation ultrafast science. in 1st paragraph of page 2.

Response: We fully stand by this sentence. As already discussed in our response to the General Opening Comment above, the characteristics of optical pulses produced using the RDW approach are highly desirable and not easily accessed by other means. RDW-based light sources are therefore set to play a leading role in shaping ultrafast science over the next decade, with several leading research groups and

high-profile central-user facilities now beginning to invest in implementing this technology – in multiple cases as part of a direct collaboration with some of authors included in this current manuscript.

Changes: None.

Comment 3: *What is the meaning electronically excited?*

Response: In this context, electronically excited refers to absorption of UV light leading to the transfer of an electron within a molecule to a more energetic, although still bound state. Essentially this causes a transition from the ground electronic state to an excited electronic state. This is commonly understood language in the framework of molecular spectroscopy – although we now realize that the phrasing could potentially be interpreted in misleading way. Given the broad scope of the Nat. Comms. readership, we appreciate this could therefore be made clearer.

Changes: We have removed the occurrences of “*electronically excited state*” on pages 2 and 4 and replaced these with the term “*photo-excited electronic states*”.

Comment 4: *Revise the sentences between line 53-56*

Response: The sentence in question is: “*The excellent time resolution afforded by this approach allows us to identify two distinct pathways governing molecular photofragmentation that have been inferred previously from static spectroscopy measurements but never observed directly.*”

Changes: This sentence has already been modified in response to Comment 5 from Reviewer #3.

Comment 5: *Revise the sentences between line 60-63*

Response: The sentence in question is: “*Our work provides a powerful demonstration of the unique capabilities afforded by RDW emission as a light source for ultrafast spectroscopy and, furthermore, showcases the current state-of-the-art synergy between advanced quantum chemistry calculations and cutting-edge experimental measurements.*”

Changes: This text has been modified to the following: “*Our work provides an important demonstration of the capabilities afforded by RDW emission as a light source for ultrafast spectroscopy and, furthermore, showcases the utility of combining advanced quantum chemistry calculations and experimental measurements.*”

Comment 6: *When the figure information is introduced in the beginning of the sentences, please use Figure 1. Fig. 1 is doable in the middle of the sentence. Please revise this issue throughout the manuscript.*

Response & Changes: This is straightforward to rectify. All instances of Fig. appearing at the start of a sentence (page 3, 6 and 10) have been amended to Figure.

Comment 7: *Please redraw all the figures with a b c d to highlight the panel. The indication of left right are not a scientific writing way.*

Response & Changes: We have amended all multi-part figures to incorporate a, b, c... labels. We have also updated the associated captions and (where relevant) any references to these figures in the main text.

Comment 8: *The spectrum of the light field is not essential in this manuscript. please plot a schematic diagram of the potential energy surface or reaction pathway in the measurement, which is more important than the spectrum for readers to understand the ultrafast dynamics happened in morpholine.*

Response: Based on previous experiences, the capillary input and output spectra are of importance and interest to anyone wanting to implement their own RDW set-up. We therefore feel that these data should remain part of Fig. 1. We don't feel that an additional schematic diagram is required given the reaction pathway plots already provided in Figs 7 and 9. It's not clear to us what extra information such a schematic diagram would provide.

Changes: None.

Comment 9: *Line 97, highly differential*

Response: The full text being referred to here is "*The TRPEI approach is a powerful pump-probe technique which offers highly differential time-, energy- and angle-resolved information in a single set of experimental measurements*". In our opinion, the combination of time-, energy- and angle-resolved data warrants the use of "highly differential" as a descriptor.

Changes: None.

Comment 10: *Line 101, as this next-generation experiment*

Response: The full text being referred to here is: "*The use of TRPEI coupled with an HCF-based RDW output represents an important advance in the field of excited state molecular dynamics as this next-generation experiment significantly improves the observation of the earliest time events after absorption of a DUV photon*".

Changes: We have replaced "*as this next-generation experiment*" with "*as our experiment*"

Comment 11: *line 104, novel spectroscopic investigations*

Response: The full text being referred to here is "*This exploits the extremely short RDW pulse duration and opens interesting avenues for novel spectroscopic investigations in the ultrafast domain.*" We stand by this statement. As already discussed earlier, the characteristics of optical pulses produced using the RDW approach are not easily accessed by other means and so exploiting this will clearly present new opportunities for novel ultrafast spectroscopy measurements moving forwards.

Changes: None.

Comment 12: *Line 105, schematic structure in Fig.2, not necessary*

Response: The full text being referred to here is: "*Morpholine (schematic structure in Fig. 2) is a saturated cyclic secondary amine that provides an excellent starting model for investigating the fundamental photophysics of the N-H chemical bond, which is ubiquitous throughout nature*". Many readers will not be immediately familiar with the morpholine molecule, and so signposting towards an instructive diagram at this point (where we first start to discuss the structure) seems entirely appropriate.

Changes: None.

Comment 13: Line 108, below ca. 255nm

Response & Changes: Amended to “below *approx.* 255 nm”

Comment 14: line 112, at excitation >220nm

Response & Changes: Amended to “at excitation *wavelengths longer than* 220 nm”

Comment 15: Line 142, what is the meaning of raw image data? Are they the measured data from MCP? If not, please use the right words to explain the data.

Response: Raw image data refers to the 2D projection image of the 3D photoelectron angular distribution acquired by a CCD camera over an extended timeframe at a given pump-probe delay. We appreciate that this definition can be made clearer.

Changes: Where we first make use of the term “raw” in relation to image data (line 130), we have re-written the text to provide the reader with more detail: “*The popular velocity-map imaging (VMI) technique²¹ was used to acquire raw TRPEI data. This comprised a set of 2D projection images of the full 3D-PAD at each pump–probe delay, recorded using a micro-channel plate and phosphor screen assembly in conjunction with a CCD camera (see Methods for more details). Polar basis-set expansion (pBASEX)²² was then used to process this data, permitting quantitative analysis of the excited state dynamics operating in morpholine following DUV excitation at around 250 nm using our RDW source.*”

Comment 16: Line 149-150, please add the intensity calibration of the pump and probe pulse

Response: This technical information is included later in the Methods section: “*Focused optical intensities at the interaction region were of order 10^{13} W cm⁻² for both the pump and the probe, yielding associated Keldysh parameters³⁸ >1. This suggests that strong-field tunnel ionization effects are not expected to be significant – as confirmed in our experimental observations (Figs. 2 and 3). Although the Keldysh parameter is known to be a limited approximation for molecular ionization,³⁹ it still serves as a useful indicative benchmark when evaluating our starting experimental conditions*”. We feel that moving this text into the main Results section will be detrimental to the overall narrative flow.

Changes: None.

Comment 17: Line 152, most of time dependent results are plotted in the format with an x-axis for pump-probe time delay, and y-axis for photoelectron distribution. The current Fig.2 is not friendly for readers to identify the time dependent evolution.

Response: The presentation format we have chosen for the 3D plots in Fig. 2 (using the x-axis for pump–probe delay) is commonly used within the ultrafast spectroscopy community. We therefore feel that switching the x- and y-axes will not make the plot any more user-friendly.

Changes: None.

Comment 18: Line 192, here, should be where

Response & Changes: This has been corrected.

Comment 19: *The most important information in Fig.4 is the oscillation in each beta. The author should discuss with the theoreticians to find a proper way to tell the story of the fast oscillation overlapping over the long-term decay.*

Response: We agree that the apparent oscillation in the anisotropy parameters presented in Fig.4 is of some interest. We comment on this in the Discussion section on page 18/19 as follows: “Furthermore, we note that the weak modulation in the various anisotropy parameters with time, as seen in Fig. 4, exhibits some resemblance to the oscillatory behaviour seen in the pyramidalization angle plot of Fig. 8. We therefore tentatively suggest that periodic evolution of the morpholine geometry in this coordinate may be responsible for these more subtle changes in overall PAD structure. More detailed Fourier analysis of this aspect of the data is, however, inconclusive”. A more detailed theoretical investigation into this behaviour (and the PADs more generally) is beyond the scope of the current work, but we do intend to return to this in future – and this is something we can highlight more explicitly.

Changes: Text on page 19 has been amended as follows: “More detailed Fourier analysis of this aspect of the data is, however, inconclusive – *although an expanded theoretical exploration of PAD anisotropy and the associated temporal evolution will form the basis of future investigations.*”

Comment 20: *Line 219. Two additional discussions should be included. One is the electronic states crossing, from the oscillation of the beta parameters, a clear illustration of the potential energy surface conical intersection should be provided. Second is the exact meaning of the 115 fs time.*

Response: This is an interesting point to raise. The oscillations in beta parameter are likely to carry information on changes in the electronic structure, for instance as the wavepacket oscillates in the vicinity of a conical intersection where the character of the adiabatic electronic states changes over time. This is what we already indicate on page 11: “A single-exponential fit to the average of these data (bottom panel of Fig. 4) reveals a 1/e lifetime τ_β of 115 ± 10 fs. This is clearly distinct from the lifetimes revealed by the global fit to the transient photoelectron signal (7 fs and 380 fs) and suggests that the electronic character of the $3s/n\sigma^*$ state may evolve differently to that of the population decay”. However, modelling this process to gain a more detailed interpretation is a nontrivial task that requires partial ionization cross sections to be calculated accurately. Furthermore, the current experiment exploits multiphoton ionization, which is even more challenging to model using current state-of-the-art methods such as R-matrix theory. Nevertheless, we fully agree that this constitutes an interesting direction, and we very much plan to explore it further in future work (as already highlighted in our response to Comment 19 above).

Changes: None.

Comment 21: *Line 246. All significant pathways have been well discussed by previous work. The results shown in this paper only present its real evolution in time scale. There is no significant observations are reported. This reduce the influence of the paper.*

Response: We disagree. The decay pathways had been speculated in a single previous work [*Chem. Sci.* **1**, 89-96 (2010)] but never observed/confirmed directly. One of the key results of our work is that we can experimentally observe these pathways, and their timescales, and then combine these observations with simulations, which combined provide a much more complete insight into the evolution of the dynamics. Furthermore, that previous work could not measure the timescales of the decay pathways or

decouple the population dynamics from the structural dynamics – something that is made possible by our use of the RDW source and is a particularly important result in the general context of ultrafast molecular dynamics (hence the title of the manuscript “*Decoupling structural molecular dynamics from excited state lifetimes using few-femtosecond ultraviolet resonant dispersive waves*”).

Changes: None.

Comment 22: *Line 267, I cannot agree with the author that the experimental results of 7/380 decay time consist well with the 30/465 fs.*

Response & Changes: This is a fair point to raise. The important points here are that we see (i) the same number of distinct processes operating and (ii) the experimental and theoretical timescales are the same order of magnitude for each pathway. To better reflect this, the phrase “*good agreement*” on page 13 has been amended “*reasonable agreement*”.

Comment 23: *Line 318. The experimental illustration doesn't understand the internal structure conversion. The experimental results only focus on the time value fitted from the results, fully lose the underlying potential surface crossing and conical intersection dynamics on different states. The experimental and theoretical authors of the manuscript should well organize the manuscript to be unit.*

Response: The full text being referred to here is: “*To link the calculations more closely to the experiment, the structural evolution of these two key internal coordinates for the ensemble of trajectories on the $S_1(3s/n\sigma^*)$ potential energy surface is shown in Fig. 8*”. This was quite a difficult comment to understand, but if the suggestion is for a wide-ranging rewrite of the manuscript in terms of how the experimental and theoretical results are presented, this is not a reasonable revision task at this stage. We feel that we have endeavoured to coherently link the theory and the experiment as much as possible in the manuscript.

Comment 24: *Line 336. Please revise the electron orbital to be partial transparent, where the molecular structure could be well identified.*

Response & Changes: We agree that our original figure lacked clarity and have updated Fig. 8 with new versions of the orbital plots that are higher resolution and have a viewing angle that is easier to interpret.

Comment 25: *The authors mentioned in the main text that they implemented several upgrades to their RDW-based laser system to extend its usable lifetime and to improve the shot-by-shot noise performance. Until to the methods, neither detailed upgrading techniques used in current set-up nor the stability-performance parameters are provided in the manuscript. I strongly suggest the authors to add a table that summarizes some parameters of both the pump and the probe pulses used in the experiment, including their energies, widths, optical bandwidths and shot-by-shot stabilities. Such a table would be very helpful for readers to quickly catch the key performance of this ultrafast laser system.*

Response: The primary focus of this paper is the application of RDW-based sources to molecular dynamics studies. As such, we feel that technical details relating to the experimental setup are best placed in a separate Methods section at the end of the manuscript (which is also standard practice for Nature Communications). This is clearly signposted on page 3: “*A full technical overview of the experimental infrastructure and optical beam paths may be found elsewhere,^{6,13} with a shorter summary also provided in the Methods section.*” The manuscript already contains 10 display items (9 figures and 1 table), which

is at the limit for Nature Communications. Another table documenting optical source properties is therefore not possible without removing something else – which we feel would weaken other aspects of manuscript. Nevertheless, we appreciate that we could improve the signposting to the technical information in the Methods section and, furthermore, add some additional useful information based on this comment.

Changes: The text on page 3 has been amended as follows: “A full technical overview of the experimental infrastructure and optical beam paths may be found elsewhere,^{6,13} with a shorter summary (including all relevant optical pulse parameters) also provided in the Methods section.”

The Figure 1 title caption on page 4 has also been improved for better signposting: “Fig. 1 | Summary of spectral parameters (see Methods section for more details).”

We have also added some more information about the pulse bandwidth in the Methods section on page 21: “This removes the need for a dispersive exit window and leads to extremely short DUV pulse durations at the HCF output (the spectrum centred at 250 nm in Fig. 1 has a FWHM bandwidth of 20.5 nm, corresponding to a Gaussian pulse duration of 4.5 fs FWHM at the Fourier transform limit).”

Decoupling structural molecular dynamics from excited state lifetimes using few-femtosecond ultraviolet resonant dispersive waves (NCOMMS-25-22152)

Response to Specific Comments (second iteration):

Reviewer #3

Comment 1: *I appreciate the experimental problems that the authors report with backlash from their stage. However, this does result in a lower than necessary sampling of the pump-probe time delay. To demonstrate the reproducibility of their IRF and qualify their additional statement, they have added to the revised manuscript, they should also show the repeat IRF measurements in the supporting information.*

Response: Currently there is no Supporting Information (SI) document for this manuscript, and the overall story is fully self-contained within the main text. As such, we feel that generating SI to accommodate a solitary additional figure is not a good way to proceed – especially as the subject matter is an aside from the main thrust of the paper's focus. We appreciate that the point raised by this reviewer in their original report relating to sampling was an important one, but we feel that the text we added in our first revision is sufficient to highlight this issue.

Changes: None – although considering this comment in detail has had input into our response to Comment 2 below.

Comment 2: *Some groups indeed choose to quote time constants below their IRF. However, in those cases, the sampling of the pump-probe time delay is higher than in the present study. Therefore, the authors should quote the time constant as IRF limited rather than 7 ± 2 fs, especially as they have carefully reworded the manuscript to indicate this is a lower bound to time constant associated with the prompt first N-H bond dissociation channel.*

Response: In considering Comment 1 above, we have re-evaluated the error in our IRF function $g(\Delta t)$ and revise this from 11 ± 1 fs to 11 ± 2 fs. This places the IRF and the τ_1 lifetime within each other's error bars. We feel that quoting 7 ± 2 fs for the latter is valid but can caveat this in the main text.

Changes: The text on page 7 has been updated to: “*These were found to have decay constants of $\tau_1 = 7 \pm 2$ fs and $\tau_2 = 380 \pm 10$ fs – with the former being extremely close to (and within uncertainty overlap) of $g(\Delta t)$.*”

Comment 3: *There are two points I would like the authors to reconsider in their response: 1) the molecular beam temperature is not characterised, and so it is unclear what the relative Boltzmann populations will be. 2) A combined experimental and theoretical study (ref. 19) determined the energy between the ground state axial and equatorial conformers to be 0.022 eV, three times lower than the calculated value here.*

Response: We are unable to characterize our molecular beam temperature but, given the expansion conditions, conservatively assume a temperature of 100 K (with the true value likely to be lower). In this scenario, the 0.022 eV energy gap quoted above would still only yield 8% of the axial conformer and so may (to a first approximation) be considered negligible in our analysis.

Changes: We now explicitly quantify our temperature estimate on page 18: “*Under the cold molecular beam expansion conditions of our experiment (assumed ≤ 100 K)...*”

Comment 4: *For clarity, the authors need to define whether they are referring to the adiabatic or diabatic S1 and S0 states. I believe it is adiabatic. If the reader is not in this mindset, it complicates reading the manuscript. Briefly stating this or reminding the reader when necessary will be helpful. Especially to clarify what is meant by “decay to the S0 state”, a phrase which is used multiple times in the manuscript.*

Response: We are referring to adiabatic states.

Changes: This is now clarified on page 4 where the first technical discussions begin: “*Note here that we refer to adiabatic states throughout*”.

The manuscript of "Decoupling structural molecular dynamics from excited state lifetimes using few-femtosecond ultraviolet resonant dispersive waves" by S. L. Jackson et al., reported the ultrafast measurement of excited state lifetime in the complex molecules of morpholine by employing a ultrashort VUV pulse generated through a upcoming RDW approach in hollow-core fiber. Techniquely, a system response function of 11fs was realized, and two distinct dissociation pathways of N-H bond fission was identified through the time-resolved photoelectron angular distribution. The authors comments that the RDW would be possible to revolutionize ultrafast science over next decade, which is overestimate the fessibility of the RDW phenomenon and associated ultrafast technique. In basic, the RDW associated ultrashort pulse even close to sub-cycle attosecond light field theoretically, its influence could not be exaggerated to revolutionizing ultrafast science, the past one who make it is the high harmonic generation and attosecond pulse generation. RDW provides an alternative approach to generate ultrashort pulse but still within the region of femtosecond and attosecond science. However, the application of RDW to perform ultrafast phenomenon beyond the ultrafast pulse generation is highly recommended. Generally, the mamuscirpt is poorly writen. A thourough revision is necessary to rethink about the publication of the manuscript including the following details.

1. Abstract

* As mentioned above, the introduction of RDW and its application in ultrafast phenomenon should be constricted to certain directions and areas.

* revise all the sentences with extremely, excellent. Acuturally, the time resolution of the ultrafast dynamics measurement is far below the approach based attosecond pulse through HHG.

* Please revise the conclusion of ... is not possible without the A VUV pulse generated through high harmonic generation could reach the same effect, by including the additional field locking methods, the time resolution could be even higher that the current pump-probe metrology in this manucript. So that a fair and humble expression is very necessary. The experimental discriptions have the same issue throughout the whole manucript, and the words suddenly return to normal status in the thoeretical part.

* The theoretical demonstration is very important in this manuscript, its role was excessively weakened in the introduction.

2. revise this sentence. ... which will play a leading role in next-generation ultrafast science. in 1st paragraph of page 2.

3. what is the meaning electronically excited?

4. revise the sentences between line 53-56

5. revise the sentences beteen line 60-63

6. When the figure information is introduced in the beginning of the sentences, please use Figure 1. Fig.1 is doable in the middle of the sentence. Please revise this issue throughout the manuscript.

7. Please redraw all the figures with a b c d to highlight the panel. The indication of left right are not a scientific writing way.

8. The spectrum of the light field is not essential in this manuscript. please plot a schematic diagram of the potential energy surface or reaction pathway in the measurement, which is more important than the spectrum for readers to understand the ultrafast dynamics happened in morpholine.

9. line 97, highly differential

10. line 101, as this next-generation experiment

11. line 104, novel spectroscopic investigations

12. line 105, schematic structure in Fig.2, not necessary

13. line 108, below ca. 255nm

14. line 112, at excitation >220nm

15. line 142, what is the meaning of raw image data? Are they the measured dat from MCP? If not, please use the right words to explain the data.

16. line 149-150, please add the intensity calibration of the pump and probe pulse

17. line 152, most of time dendent results are plotted in the format with an x-axis for pump-probe time delay, and y-axis for photoelectron distribution. The current Fig.2 is not friendly for readers to identify the time dependent evolution.

18. line 192, here, should be where

19. The most important information in Fig.4 is the oscillation in each beta. The author should discussed with the theoreticians to find a proper way to tell the story of the fast ocillation overlapping over the long-term decay.

20. line 219. Two additional discussions should be included. One is the electronic states crossing, from the osciallation of the beta parmeters, a clear illustration of the potential energy surface conical intersection should be provided. Second is the exact meaning of the 115 fs time.

21. line 246. All significant pathways have been well discussed by previous work. The results shown in this paper only present its real evolution in time scale. There is no significant observations are reported. This reduce the influence of the paper.

22. line 267, I cannot agree with the author that the experimental results of 7/380 decay time consist well with the 30/465 fs.

23. line 318. The expeirmental illustration doesn't understand the internal structure conversion. The experimental results only focus on the time value fitted from the reuslts,

fully lose the underlying potential surface crossing and conical intersection dynamics on different states. The experimental and theoretical authors of the manuscript should well organize the manuscript to be unit.

24. line 336. Please revise the electron orbital to be partial transparent, where the molecular structure could be well identified.

25. The authors mentioned in the main text that they implemented several upgrades to their RDW-based laser system to extend its usable lifetime and to improve the shot-by-shot noise performance. Until to the methods, neither detailed upgrading techniques used in current set-up nor the stability-performance parameters are provided in the manuscript. I strongly suggest the authors to add a table that summarizes some parameters of both the pump and the probe pulses used in the experiment, including their energies, widths, optical bandwidths and shot-by-shot stabilities. Such a table would be very helpful for readers to quickly catch the key performance of this ultrafast laser system.

In summary, the current manuscript cannot be published in Nature communications and even else where. A scientific writing and complete revision is very necessary.